# Inactivation of KCNQ1 potassium channels reveals dynamic coupling between voltage sensing and pore opening

Panpan Hou[1], Jodene Eldstrom[2], Jingyi Shi[1], Ling Zhong[1], Kelli McFarland[1], Yuan Gao [3], David Fedida[2] & Jianmin Cui[1]

In voltage-activated ion channels, voltage sensor (VSD) activation induces pore opening via VSD-pore coupling. Previous studies show that the pore in KCNQ1 channels opens when the VSD activates to both intermediate and fully activated states, resulting in the intermediate open (IO) and activated open (AO) states, respectively. It is also well known that accompanying KCNQ1 channel opening, the ionic current is suppressed by a rapid process called inactivation. Here we show that inactivation of KCNQ1 channels derives from the different mechanisms of the VSD-pore coupling that lead to the IO and AO states, respectively. When the VSD activates from the intermediate state to the activated state, the VSD-pore coupling has less efficacy in opening the pore, producing inactivation. These results indicate that different mechanisms, other than the canonical VSD-pore coupling, are at work in voltage-dependent ion channel activation.

[1] Department of Biomedical Engineering, Center for the Investigation of Membrane Excitability Diseases, Cardiac Bioelectricity and Arrhythmia Center, Washington University in St Louis, St Louis, MO 63130, USA. [2] Department of Anesthesiology, Pharmacology and Therapeutics, University of British Columbia, Vancouver, British Columbia, Canada V6T1Z3. [3] Tencent AI Lab, Shenzhen 518052, China. Correspondence and requests for materials should be addressed to J.C. (email: jcui@wustl.edu)

Voltage-activated ion channels contain functionally and structurally distinctive voltage sensor domains (VSDs), comprised of transmembrane helices S1–S4, and a central pore formed by the S5–S6 structures from four subunits or repeated domains[1–5]. Upon membrane depolarization, charged residues in the VSD sense the membrane's electric field change and initiate conformational changes for voltage sensor activation[1–3, 6, 7]. These movements in the VSD are propagated to the pore via interactions between the VSD and the pore to open the channel for ion permeation across the membrane[8, 9]. A canonical mechanism for the VSD-pore coupling proposes that the interactions between the S4–S5 linker and the cytosolic end of S6 primarily link VSD movements and pore opening[10–14].

**Fig. 1** Time and voltage dependence of KCNQ1 hook currents. **a** Left, representative KCNQ1 fast decay current recorded in ND96 solution from a triple pulse protocol. The hyperpolarizing pulse was for 20 ms at −120 mV, and the depolarizing pulse was to +40 mV. The inset shows the fast decaying currents in response to the return of the voltage to +40 mV after hyperpolarization, with expanded current and time scales. Right, representative KCNQ1 + KCNE1 current recorded under the same conditions. **b** Left, time dependence of KCNQ1 hook currents recorded in high potassium (100 mM K$^+$) solution. The pre-pulses were +40 mV with time durations ranging between 0.02 and 4.355 s, and the test pulse was 2 s long at −120 mV. The inset shows the hook in tail currents with an expanded time scale. Right, representative KCNQ1 + KCNE1 current recorded under the same conditions. **c** Voltage dependence of KCNQ1 hook currents recorded in high potassium (100 K$^+$) solution. The pre-pulse was 4 s with voltages ranging from −80 to +60 mV, and the test pulse was 2 s at −120 mV. The inset shows tail currents with an expanded time scale. **d** KCNQ1 hook current (black) fitted with the double exponential function $F(t) = A_1 \times \exp(-t/\tau_1) + A_2 \times \exp(-t/\tau_2) + C$. The fitting curve, slow component ($A_1 \times \exp(-t/\tau_1) + C$), and hook component ($A_2 \times \exp(-t/\tau_2)$) are shown in red, cyan, and blue, respectively. **e** Normalized $A_1$ (top) and $A_2$ (bottom) of KCNQ1 hook currents vs. time durations of the pre-pulse, with voltages ranging from −60 to +60 mV ($n \geq 4$). Error bars are SEM

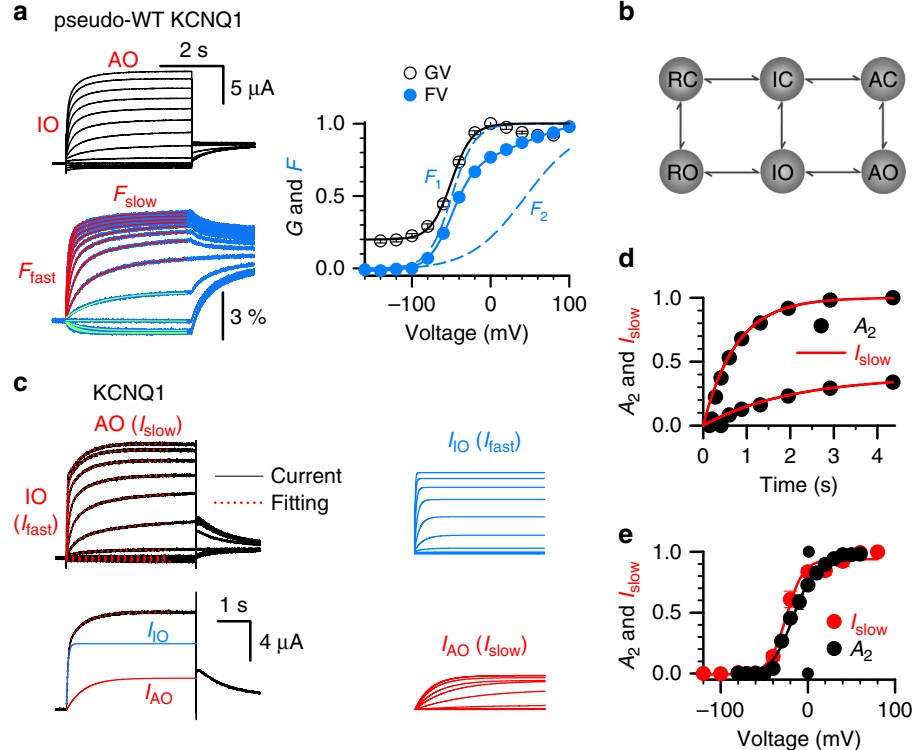

**Fig. 2** Similar time and voltage dependence of hook currents and the AO state of the KCNQ1 channel. **a** VCF recordings of pseudo-WT (C214A/G219C/C331A) KCNQ1 currents (black) and fluorescence (blue), with voltages from −140–100 mV in 20 mV increments and back to −40 mV to test the tail currents. The fluorescent traces were fitted with a single (green) exponential equation below −50 mV, and a double (red) exponential equation above −50 mV showing both $F_{fast}$ and $F_{slow}$ components. Right panel, the G–V (black, $V_{50} = −49.12$ mV, slope = 13.81) and F–V (blue) relationships with $F_1$ ($V_{50} = −49.17$ mV, slope = 14.67) and $F_2$ ($V_{50} = 43.57$ mV, slope = 38.00) components (dotted lines). **b** The scheme shows the two-step VSD movements R ↔ I ↔ A and channel opening C ↔ O. **c** Upper left, the WT KCNQ1 currents (black lines) recorded with voltage ranging from −120–80 mV were fitted with a double exponential function (red dots). The tail currents were measured at −40 mV. Lower left, the current (black line) recorded at 40 mV was fit with a double exponential function (red dots). The blue trace is the fast component ($I_{fast}$) representing the current of the IO state, and the red trace is the slow component ($I_{slow}$) representing the current of the AO state. Right, $I_{fast}$ and $I_{slow}$ from the currents in upper left. **d** Dependence of $A_2$ on the pre-pulse duration (black, from Fig. 1e) and $I_{slow}$ on test pulse duration (red, from Fig. 2c) at −20 and +40 mV. **e** The dependences of $A_2$ on pre-pulse voltages (black, from Fig. 1e) and $I_{slow}$ on test pulse voltages (red, from Fig. 2c) of KCNQ1 channels. The voltage for half activation ($V_{1/2}$) of $A_2$-V and $I_{slow}$-V is −18.2 ± 2.1 mV and −25.0 ± 2.7 mV, respectively. $n \geq 4$. Error bars are the SEM. RC resting closed, IC intermediate closed, AC activated closed, RO resting open, IO intermediate open, AO activated open

KCNQ1, also known as Kv7.1 or KvLQT1, belongs to a subfamily of voltage-activated K$^+$ (Kv) channels[15–21]. Our recent studies show that, similar to other Kv channels, the VSD of KCNQ1 activates in resolvable steps: from the resting state to an intermediate state, and then to the activated state. However, unlike some other Kv channels that open only when the VSD is activated, the KCNQ1 is open at both the intermediate and activated VSD states, resulting in the intermediate open (IO) and activated open (AO) states[22, 23]. The two open states exhibit different ionic permeation, PIP$_2$ modulation, and pharmacological properties, and a mutation in a conserved site for the interactions between the S4–S5 linker and the cytosolic end of S6 specifically suppresses the IO state[22, 23]. These results suggest that the VSD-pore coupling has different properties in the IO and AO states. However, it remains unclear if the same canonical VSD-pore coupling underlies both the IO and AO openings with some minor modifications, or if distinctive mechanisms are responsible for the VSD-pore coupling in the two open states. KCNQ1 is expressed in cardiac myocytes, epithelial cells, and neurons, where it co-assembles with various members of the auxiliary KCNE subunit family and exhibits drastically different phenotypes[24]. The function of these channels is vital to physiological processes associated with the tissue type, and mutations and polymorphisms in the *KCNQ1* gene are linked to cardiac

arrhythmias, congenital deafness, and type II diabetes[15, 25–27]. The different functional properties of these channels are at least in part due to differential modulations of the IO and AO states by KCNE subunits[22]. Therefore, to understand VSD-pore coupling in the IO and AO states is important for understanding the basic principles of voltage-dependent ion channel activation as well as how KCNQ1 controls various physiological processes.

Kv channels are activated by membrane depolarization, but many of these channels spontaneously close after opening though the membrane remains depolarized. This spontaneous closing of the channel is known as inactivation and is a fundamental property of voltage gated ion channels. Various molecular mechanisms for inactivation have been revealed in previous studies[28–30]. Fast N-type inactivation occurs via a "ball and chain" mechanism that involves a cytoplasmic peptide domain at the N-terminus of the α- or β-subunit, which occludes the channel central cavity and prevents ion permeation[28]. Slow C-type inactivation is suggested to involve structural rearrangements in the outer pore, leading to a loss of K$^+$ coordination sites in the selectivity filter[28–30]. In these mechanisms, inactivation is a conformational state of the channel protein that is non-conducting and different from an open state[29, 30]. The KCNQ1 channel undergoes activation as well as partial inactivation, which is evident from the fast decaying current following

a brief hyperpolarization during a depolarizing voltage pulse (Fig. 1a). The brief hyperpolarization allows the channels to recover from inactivation, and then the channels rapidly re-inactivate when the depolarization is resumed[31–34]. However, the inactivation of KCNQ1 does not exhibit the hallmarks of canonical N-[28, 35] or C-type inactivation[28–30], and the mechanism underlying this inactivation remains elusive.

In this study, we find that the inactivation in KCNQ1 channels derives from the different mechanisms of the VSD-pore coupling that lead to the IO and AO states, respectively. A comparison of the inactivation and the changes in the IO and AO states shows that the two processes coincide in their time course and voltage dependence. Mutations that alter the VSD-pore coupling in AO or IO states also similarly alter the inactivation. Single-channel recordings and kinetic analyses of VSD movements and channel openings indicate that the VSD-pore coupling in the AO state results in a lower open probability than in IO, so that the transitions from IO to AO give rise to the inactivation-like behavior. These results not only reveal a different mechanism for the inactivation in KCNQ1 channels, but also indicate distinctive mechanisms for VSD-pore coupling in the IO and AO states.

## Results

**Correlations between inactivation and IO to AO transitions.** During a depolarizing voltage pulse that activates KCNQ1 channels, a fast decaying current following a brief hyperpolarization has been typically viewed as evidence of KCNQ1 channel inactivation. A brief hyperpolarization (20 ms at −120 mV) allows the channels to recover from inactivation and then rapidly re-inactivate when the depolarization is resumed[31–34] (Fig. 1a). Inactivation is also seen as a hook in the tail currents, i.e., an initial increase of the inward (downward) tail current followed by continuous decay of the current upon repolarization after a pre-pulse that activates KCNQ1 channels (Fig. 1b). This hook current is consistent with the idea that channels recover from inactivation upon repolarization via an open state before closing due to deactivation[33, 34, 36, 37]. When coexpressed with the KCNE1 subunit, $I_{Ks}$ (KCNQ1 + KCNE1) channels no longer show the fast decaying current or hook current (Figs 1a, b), indicating that KCNE1 association eliminates the inactivation phenotype[31, 32, 36, 37].

To resolve the gating processes that lead to hook currents, we recorded tail currents during hyperpolarization to −120 mV, after varying the duration and potential of pre-pulses that activated KCNQ1 channels (Fig. 1b, left, Fig. 1c). It is obvious that the hook developed with time and voltage. With longer pre-pulse time durations, the hook grew bigger, and with higher pre-pulse voltages, the hook grew bigger as well (Fig. 1b, left, Fig. 1c). To quantify the KCNQ1 inactivation, we fitted the tail currents with a double exponential function $F(t) = A_1 \times \exp(-t/\tau_1) + A_2 \times \exp(-t/\tau_2) + C$ (Fig. 1d)[36]. Here, $A_1$ and $A_2$ are the amplitudes, $\tau_1$ and $\tau_2$ are the time constants, and $C$ is the offset of the leak current. A representative tail current (black), recorded after a pre-pulse of 4 s duration at +40 mV, is shown well fitted by a double exponential function (red). $A_1 \times \exp(-t/\tau_1) + C$ represents the relatively slow deactivation process (cyan dots, component 1), and $A_2 \times \exp(-t/\tau_2)$ represents the increasing inward (downward) current (blue dots, component 2). The dependence of the normalized $A_1$ and $A_2$ on the time and voltage of the pre-pulse are shown in Fig. 1e. The increasing inward current (component 2) was the cause of the "hook" in tail currents, and we use $A_2$ to quantify the inactivation in this study. It was clear that $A_2$ did not show up until the pre-pulse was above −40 mV (Fig. 1e), which suggests that the inactivation phenotype, as indicated by the hook current, did not appear when the KCNQ1 channels were activated

at low voltages, and appeared only when the channels were activated by voltages above −40 mV[36]. $A_2$ also increased with the duration and voltage of the pre-pulse (Fig. 1e), suggesting a dependence of inactivation on the activation processes of the channel during the pre-pulse.

The properties of KCNQ1 activation in Fig. 2a recapitulate our recent findings that the channel can open when the VSD is at the intermediate state and the activated state, resulting in two open states, IO and AO[22, 23]. Voltage-clamp fluorometry (VCF) was used to monitor VSD movement and pore opening simultaneously[14, 16]. In our VCF experiments, a fluorophore (Alexa 488) attached to the top of S4 of pseudo-wild type (WT) KCNQ1 (KCNQ1-C214A/G219C/C331A) reported conformational changes of the VSD activation (Fig. 2a, blue), while the ionic currents reported pore opening[14, 16, 22, 38] (Fig. 2a, black). The F–V relation could be well fitted by a double Boltzmann function with two components, $F_1$ and $F_2$ (Fig. 2a), which report the voltage dependence of VSD activation to the intermediate and activated states, respectively. $F_1$–V and $F_2$–V relations cover different voltage ranges, with $F_2$ starting at around −50 mV, indicating that the VSD moves to the activated state at voltages more positive than −50 mV. The pore of the KCNQ1 channel is open at both intermediate and activated VSD states, as indicated by the G–V relation (Fig. 2a), generating the IO and AO states[22, 23]. The time course of the fluorescence signal, which measures VSD movements, also shows two components, a fast component approximating the transition from the resting state to the intermediate state, and a slow component approximating the transition from the intermediate state to the activated state[22]. These results are also illustrated in Fig. 2a, in which representative fluorescence traces recorded at 20 mV increments are fitted with a double exponential function (red), with fast ($F_{fast}$) and slow ($F_{slow}$) components (Fig. 2a). Interestingly, the double exponential fit was needed only for the fluorescence traces recorded above −50 mV (from −40 mV to +100 mV), while the fluorescence traces recorded below −50 mV (from −140 mV to −60 mV) could be well fitted by a single exponential function (Fig. 2a, green). This result also suggests that the VSD moves to the activated state at voltages more positive than −50 mV. Figure 2b shows the simplest kinetic model that describes the two-step activation gating mechanism of KCNQ1. The pore can close and open when the VSD is in the resting, intermediate, and activated state, shown as RC, RO, IC, IO, AC, and AO, respectively. Of note, even when the VSD is at the resting state, a very small fraction of the channels can still open (RO)[39]. Since such openings are independent of voltage and rare, we will not study the RO state here.

Our VCF results indicate that the AO state of the KCNQ1 channel also develops with time and voltage. We fitted the WT KCNQ1 current traces with a double exponential function and used the fast and slow components to approximate the currents of the IO and AO state (Fig. 2c). The red traces in Fig. 2c show the time and voltage dependence of the slow component. Strikingly, the normalized $A_2$ of the hook current (black circles) and the currents of the AO state ($I_{slow}$, red lines) at different voltages, such as −20 mV and +40 mV, have overlapping time courses (Fig. 2d). Likewise, $A_2$ (black circles) and $I_{slow}$ (red circles) show very similar voltage dependence (Fig. 2e). The close correlation between the time- and voltage-dependent development of the hook current and the AO state suggests that the appearance of the inactivation phenotype closely relates to the transitions between IO (IC/IO transition) and AO (AC/AO transition) states. wild type

**Mutations altering IO or AO state also alter inactivation.** To further examine the molecular mechanisms for the closed-open transitions IC/IO and AC/AO and if these transitions produce the

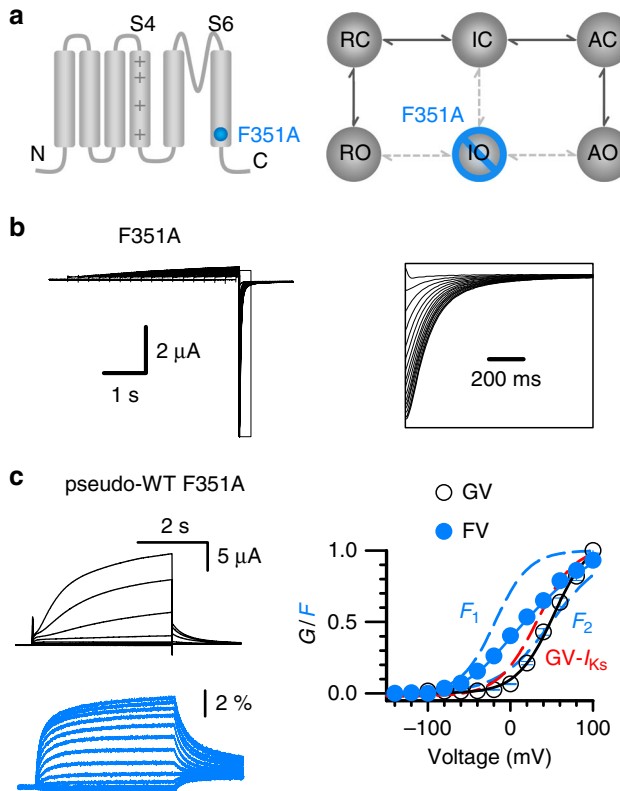

**Fig. 3** F351A suppresses the IO state and eliminates hook currents. **a** Schemes to illustrate the location of the mutation in the KCNQ1 subunit (left) and show that the IO state was suppressed by F351A mutation (right). **b** Representative hook currents of F351A recorded in 100 mM $K^+$ solution. The pre-pulses were +40 mV, with 0.02–4 s time durations, and the test pulse was 2 s long at −120 mV. The right panel shows tail currents with an expanded time scale. **c** VCF recordings of F351A currents (black) and florescence (blue). Right panel shows the $G$–$V$ (open circle) and $F$–$V$ (solid circle) relationships of F351A with the $F_1$ and $F_2$ components (dotted blue lines). The red dotted line is the $G$–$V$ relationship of $I_{Ks}$ channels. $n \geq 4$. Error bars are the SEM

inactivation in KCNQ1, we studied mutations that alter inactivation. These studies suggest that different mechanisms are responsible for the VSD-pore coupling in the IO and AO states, and the switch between these different coupling mechanisms produces inactivation.

Previous studies have shown that the KCNQ1 F351A mutation (Fig. 3a) has properties similar to those of KCNQ1 + KCNE1[22, 40], including the elimination of inactivation (Fig. 3b). F351A channels opened to the AO state with a voltage dependence shifted to more positive voltages that correlates with the $F_2$–$V$ relationship and slower activation kinetics[22] (Fig. 3c). F351A did not affect the VSD activation to the intermediate or activated state (Fig. 3c), indicating that the mutation specifically suppressed the IO state[22]. These results support the idea that the inactivation in KCNQ1 derives from both IO and AO states, the suppression of the IO state eliminates the inactivation. It was shown that KCNE1 association with KCNQ1 also selectively suppresses the IO state[22, 40], and thus this mechanism answers a long-standing question of why $I_{Ks}$ channels do not show inactivation-like KCNQ1 (Fig. 1a, b).

F351 is located in the cytosolic end of S6 and highly conserved among Kv channels. The interactions of this part of S6 with the S4–S5 linker are critical for the VSD-pore coupling in Kv channels[12, 41–43]. It is interesting that the interruption of such a

canonical VSD-pore coupling mechanism specifically suppresses IO but leaves AO almost unaltered (Fig. 3c). Does the AO state result from a different mechanism of VSD-pore coupling? Can other mutations specifically suppress the AO state? It has been shown that mutations in KCNQ1 that decrease $I_{Ks}$ (KCNQ1 + KCNE1) currents are associated with long QT syndrome (LQT)[15, 23]. Since the $I_{Ks}$ channel opens only to the AO state[22, 40], some LQT-associated mutations may alter the AO state. Therefore, to answer the above questions, we inspected the effects of several LQT-associated mutations on inactivation, based on the idea that if the mutations suppress the AO state and thus the AC/AO transitions, the inactivation of the mutant channels will be eliminated.

One of these mutations, S338F[44], was found to eliminate inactivation of KCNQ1 (Fig. 4a). Interestingly, S338F KCNQ1 expressed in the plasma membrane with a similar current size as WT KCNQ1, but unlike the WT, S338F currents could be well fit with a single exponential function (Fig. 4c), and the current was severely depressed when coexpressed with KCNE1[44] (Fig. 4c). Western blot and imaging data indicated that S338F + KCNE1 channels traffic normally to the plasma membrane[44]. This phenotype suggests that S338F might eliminate the AO state, and so the mutant KCNQ1 can reach the dominant IO state and exhibit similar current amplitudes to that of the WT. Upon KCNE1 association, the IO state was suppressed, leaving no open state in the mutant KCNQ1 + KCNE1 channels (Fig. 4c). To verify this mechanism, we made the S338F mutation in the background of the E160R/R237E mutant (E1R/R4E), which arrests the VSD in the activated state and the channel in the AO state (Fig. 4d), and F351A, which suppresses the IO state[22] (Fig. 3 and Fig. 4e). Both of these background mutant channels could open only in the AO state, and adding the S338F mutation prevented these channels from opening (Fig. 4d, e). The loss of current was due to suppression of the AO state, since the S338F/F351A channels showed normal expression in the plasma membrane, as detected by VSD movements in VCF experiments (Fig. 4e). On the other hand, adding the S338F mutation in another mutant background E160R/R231E (E1R/R2E), which arrests the VSD in the intermediate state and the channel in the IO state[7, 22, 45], showed the same normal constitutively open currents as E1R/R2E (Fig. 4d). These results support the idea that S338F specifically eliminated the AO state. Furthermore, VCF measurements showed two components in the $F$–$V$ relation of S338F KCNQ1, which are similar to those of the WT KCNQ1, while the $G$–$V$ relation simply followed the $F_1$–$V$, indicating a selectively disrupted VSD-pore coupling at the AO state (Fig. 4f, g).

We found that another LQT-associated mutation, D242N[46], shifted the voltage dependence of inactivation to more positive voltages by 36 mV (for the $A_2$–$V$ relation, $V_{1/2} = -16.7$ mV for WT and 19.5 mV for D242N) (Fig. 5). We fit the activation currents of D242N with a double exponential function and used the slow component to approximate the currents in the AO state. Compared to the WT, D242N shifted the voltage-dependent activation of the AO state by +30 mV, similar to the shift of the voltage dependence of inactivation (Fig. 5c). To verify that the mutation shifted the voltage-dependent activation to AO, we compared the shift of the $G$–$V$ relations of KCNQ1 and KCNQ1 + KCNE1 by the D242N mutation. Previous studies have shown that KCNQ1 predominantly opens in the IO state, while coexpression of KCNE1 suppresses the IO state so that the channel opens only to the AO state[22]. Compared to the WT, the $G$–$V$ relation of D242N + KCNE1 shifted 50 mV more positive, while the $G$–$V$ of D242N alone shifted only 10 mV (Supplementary Fig. 1), suggesting that D242N selectively shifted the voltage-dependent activation of the AO state. VCF measurements of the

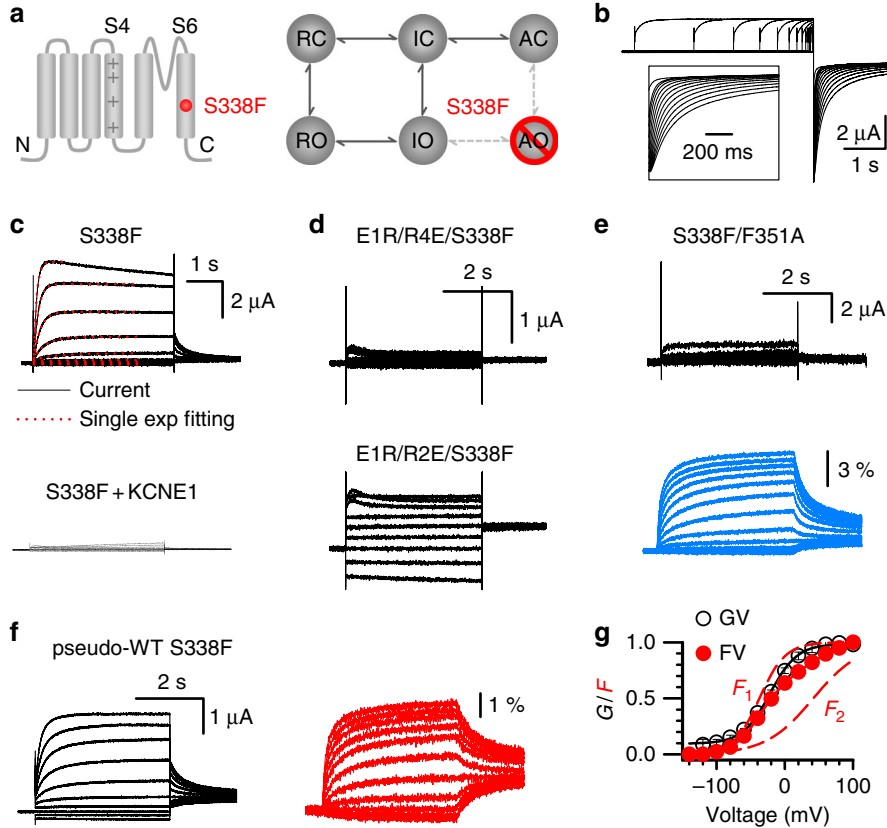

**Fig. 4** S338F suppresses the AO state and eliminates hook currents. **a** Schemes to illustrate that the AO state was suppressed by S338F mutation. **b** Representative currents of S338F recorded in 100 mM K⁺ solution. The pre-pulses were +40 mV with 0.02–4 s time durations, and the test pulse was 2 s long at −120 mV. The inset shows tail currents with an expanded time scale. **c** Representative currents of S338F and S338F + KCNE1 with voltages ranging from −100 to +60 mV. S338F currents were fitted with a single exponential function (red dots). At 60 mV, only the first 1 s of the current was fitted. **d** Representative currents of E1R/R4E/S338F and E1R/R2E/S338F with voltages ranging from −100 to +40 mV. Currents are shown on the same scale. **e** VCF recordings of S338F/F351A currents (black) and fluorescence changes (blue). **f** VCF recordings of S338F currents (black) and fluorescence (red). **g** The $G$–$V$ (black) and $F$–$V$ (red) relationships of S338F with the $F_1$ and $F_2$ components (dotted red lines). $n \geq 4$. Error bars are the SEM

D242N mutant channel showed that the $F$–$V$ relationship could be fitted with two components, with $F_1$–$V$ superimposed with that of KCNQ1, suggesting that the voltage-dependent activation of the intermediate state was not altered by the mutation. However, the $F_2$–$V$ shifted 30 mV to more negative voltages, opposite to the shift of the AO state (Fig. 5d). Since D242N did not alter $F_1$–$V$, it is unlikely that the mutation directly altered VSD movements. Therefore, the result of such opposite changes in $F_2$–$V$ and $G$–$V$ of the AO state suggests that the pore is opened less efficiently and requires higher voltages to open, while the VSD has less load during activation and requires lower voltages to activate, which means that D242N decreases the VSD-pore coupling selectively at the AO state[47]. Taken together, the results of mutations F351A (Fig. 3), S338F (Fig. 4), and D242N (Fig. 5) reveal that VSD-pore coupling in the IO and AO states is different and can be altered separately by distinctive mutations. Alterations of VSD-pore coupling by these mutations are manifested in altered inactivation, correlating to the altered IO or AO state.

**The VSD-pore coupling switch from IO to AO reduces $P_{open}$.** The above results suggest that when the VSD activates from the intermediate state to the activated state, different VSD-pore coupling mechanisms lead to the IO and AO states. The switch between these mechanisms produces the inactivation phenotype. For this to happen, the total K⁺ current through the AO state must be smaller than that through the IO state. Thus, the

inactivation in KCNQ1 would predict that either the AO state has a smaller single-channel conductance or the VSD-pore coupling for AO results in a lower open probability than that for IO. The two KCNQ1 mutations, E1R/R2E and E1R/R4E, that make the channel constitutively activated at the IO and AO state, respectively[22, 45, 48], provided an excellent tool for us to study the IO and AO states at the single-channel level (Fig. 6).

For the E1R/R2E and the E1R/R4E channels, although the macroscopic current did not show voltage or time dependence, which suggested that the channels were constitutively active, single-channel recordings showed clear transitions between open and closed states, with a relatively high open probability during active sweeps (Fig. 6a, c). Channel openings in both constructs occurred in long-lasting bursts, separated by sometimes quite long closings. Channel activity was also cyclical over the medium term, so that both constructs underwent significant periods of pulsing without activity, as exemplified by the blank sweeps at the bottom of each set in Fig. 6a, c. Both the E1R/R2E and E1R/R4E channels showed similarly small amplitude openings at +60 mV, and all-points amplitude histograms revealed mean open amplitudes for both channel constructs of about 0.025 pA at this potential, which corresponded to a single-channel conductance of ~0.18 pS (Fig. 6). The KCNQ1 blocker, chromanol 293B, blocked the single-channel currents effectively, as shown in Fig. 6b. Co-expression of the E1R/R4E mutant channel with KCNE1 increased the amplitude of single-channel openings 10-fold (~1.9 pS) as shown in the histogram, indicating that KCNE1

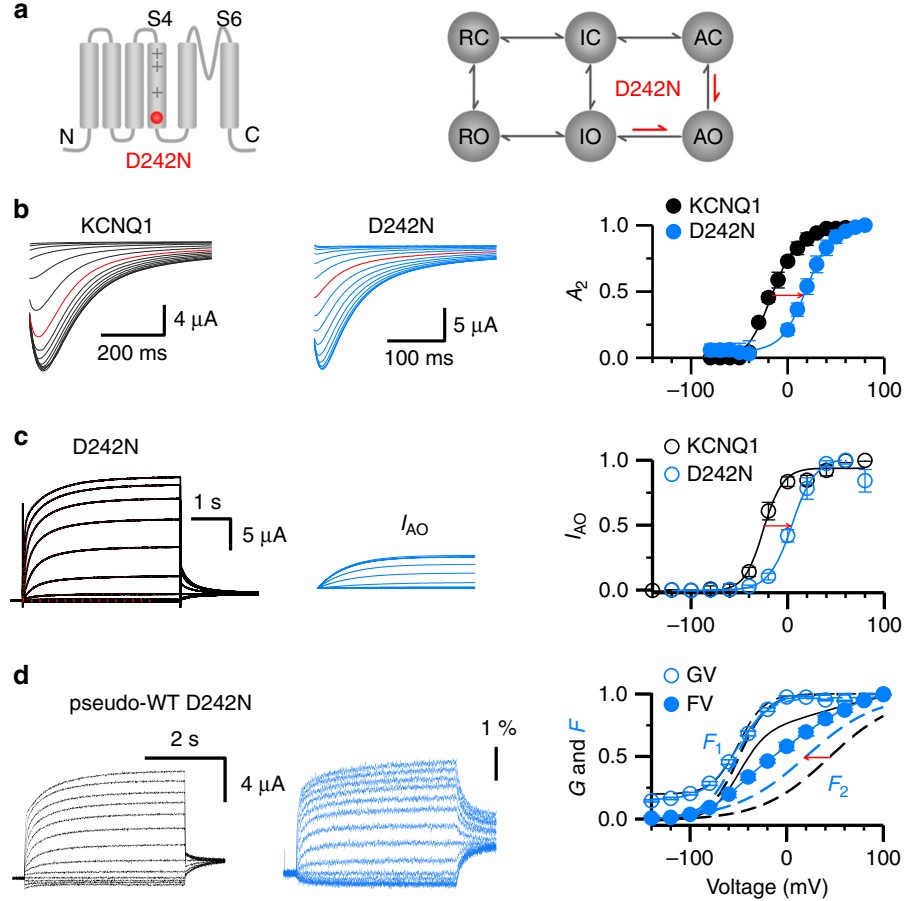

**Fig. 5** D242N shifts the voltage dependence of the AO state and hook currents. **a** Schemes showing that the D242N mutation shifted the voltage-dependent transitions of the AO state and the VSD-pore coupling in the AO state. **b** Left, the hook currents of KCNQ1 (black) and D242N (blue) recorded at −120 mV in 100 mM $K^+$ solution, with pre-pulse voltages ranging from −80 mV to +80 mV. Currents at −20 mV are shown in red for comparison. Right, hook current ($A_2$) voltage dependences of WT KCNQ1 (black) and D242N (blue). **c** Left, D242N currents (left, black lines) responding to voltages from −120 to 80 mV are fitted with a double exponential function (red dots), and the currents of the AO state ($I_{AO}$) are shown in blue (middle). Right, voltage dependence of the AO state in WT KCNQ1 (black) and D242N (blue). **d** VCF recordings of D242N currents (black) and fluorescence (red). Right panel, the $G$–$V$ (blue open circle) and $F$–$V$ (blue solid circle) relationships of D242N with the $F_1$ and $F_2$ components (dotted blue lines). The $G$–$V$ (black short dashed line) and $F$–$V$ (black line) relationships with the $F_1$ and $F_2$ components (dotted black lines) of the pseudo-WT KCNQ1 are shown in black for comparison. $n \geq 4$. Error bars are the SEM

increases single-channel conductance of the AO state in addition to suppressing the IO state[22, 49–52]. Still, although the construct was constitutively activated, it clearly retained the kinetic behavior observed with E1R/R4E alone: long bursts of activity separated by distinct quiescent periods during 4 s depolarization pulses (Fig. 6d).

Since the E1R/R2E and E1R/R4E mutants are constitutively activated, we did not compare their open probabilities, but rather examined the open probability of KCNQ1 alone during 4 s depolarization pulses to +60 mV. The individual current sweeps shown in Fig. 6e suggest that the single-channel kinetic behavior of KCNQ1 is not markedly different from that of the constitutively activated mutants. Frequent bursts of activity were separated by closed periods, and the open amplitudes were 0.023 pA (Fig. 6e), close to those of the E1R/R2E and E1R/R4E channels (Fig. 6a, c). However, we did observe that KCNQ1 channels often opened early and then showed less activity later on during depolarizing pulses. This observation is borne out in the ensemble average of single-channel currents from active sweeps (Fig. 6f), which reaches a peak amplitude early in the record close to the single-channel opening amplitude (corresponding to a mean Po of ~0.7), and then declines toward the latter part of the ensemble

trace, reaching close to an average Po of ~0.3. This change of Po with time in a pulse is much smaller and hard to discern when blank sweeps are also included in the ensemble average, which is more similar to the macroscopic current traces. The data demonstrate that KCNQ1 transitions from a higher to a lower open probability state during depolarizing pulses, with indistinguishable single-channel conductances and generally similar single-channel kinetics. These results are well represented by the switch of IC/IO transitions (IO state) to AC/AO (AO state) with VSD activation from the intermediate state to the activated state, suggesting that the VSD-pore coupling in the AO state is less efficient in opening the pore than that in the IO state. Thus, a population of KCNQ1 channels may show a smaller total current upon the transitions from the IO to AO states during a depolarization, giving rise to the inactivation phenotype.

## Discussion

The unique inactivation of the KCNQ1 channel has been a long-standing puzzle. The lack of an N-terminal ball structure within the channel indicates that KCNQ1 inactivation does not happen through the "ball and chain" mechanism[28, 35], and the lack of

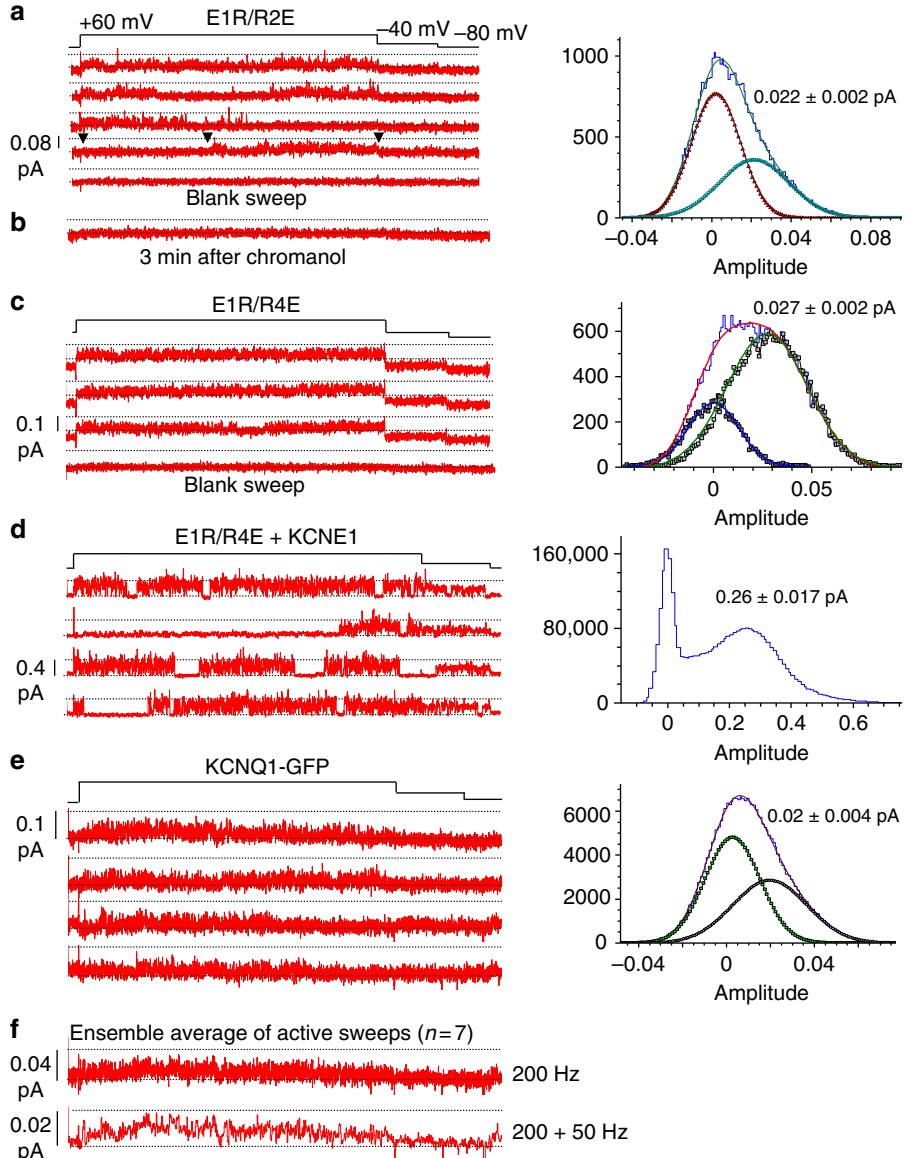

**Fig. 6** E1R/R2E, E1R/R4E, and KCNQ1-GFP single-channel recordings. The voltage protocol was the same for all recordings in this figure. The potential was stepped from a holding potential of −80 to +60 mV for 4 s, and then to −40 mV for 0.75 s, as shown above the current traces. **a** Representative single-channel current recordings made from membrane patches containing a single E1R/R2E channel. **b** Addition of 50 μM Chromanol 293B silenced all E1R/R2E channel activity after ~3 min of exposure. All-points histogram (blue) of the +60 mV portion of the fourth sweep down in **a**, with closed (maroon) and open (teal) Gaussian fits. The peak of the open component is 0.022 ± 0.002 pA. **c, d** Representative recordings of a single E1R/R4E channel (**c**) or E1R/R4E + KCNE1 (**d**). All-points histogram (blue) of E1R/R4E of the +60 mV portion of a sweep, showing a significant closed portion, with closed (blue) and open (green) Gaussian fits (**c**) and of 25 active E1R/R4E + KCNE1 sweeps (**d**). **e** Representative traces of single-channel recordings made from membrane patches containing a single GFP-tagged KCNQ1 channel. All points histogram (blue) of the +60 mV portion of sweeps in **e**, with closed (green) and open (gray) Gaussian fits. The peak of the open component is 0.02 ± 0.004 pA. **f** Ensemble average of seven active sweeps (of 75 total) of KCNQ1-GFP. The ensemble average was filtered again at 50 Hz to show the current waveform more clearly

inhibition by a high external K$^+$ concentration suggests that it is different from C-type inactivation[29, 30, 53, 54]. On the other hand, KCNQ1 shows fast and partial inactivation, which is evident only at certain voltage and time conditions. For instance, the hook current and fast decaying current do not appear until the potential exceeds −50 mV (Figs. 1 and 2), and even at +40 mV depolarization, inactivation does not appear within the first 100 ms[36] (Fig. 1). These phenomena could be well explained by the property that the KCNQ1 channel opens when the voltage sensor activates to both an intermediate and the activated state, resulting in the IO and AO states[22]. Our study indicates that the AO state is coupled to the activated state of the VSD, while the IO state is

coupled to the intermediate state of the VSD, different mechanisms of VSD-pore coupling lead to the IO and AO states, respectively, with the VSD-pore coupling being less efficient in opening the pore in the AO state (Figs. 2–6). Thus the inactivation in KCNQ1 is the manifestation of the switch in different coupling mechanisms between VSD and the pore, and the study of inactivation reveals the properties of the two coupling mechanisms.

Presented here is a kinetic model based on the above suggested mechanisms that can recapitulate all the activation and inactivation characteristics of KCNQ1, including the hook current at the early phase of the tail current after depolarization and the fast

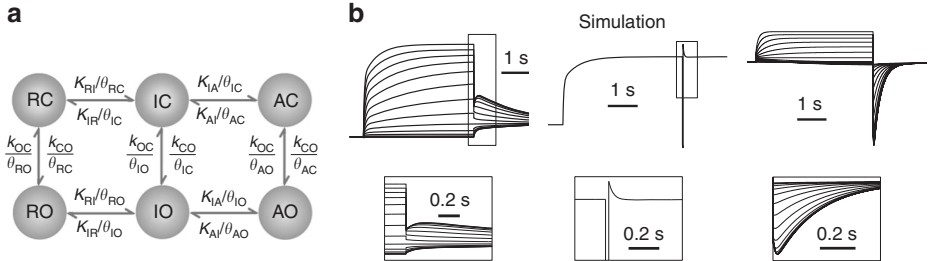

**Fig. 7** A kinetic model accounts for the inactivation phenotype in KCNQ1. **a** Kinetic model of KCNQ1 channel gating. $K_{ij} = k_{ij} \times \exp(z_{ij} \times F \times V/(R \times T))$, where $K_{ij}$ is the voltage-dependent rate of transition from VSD state $i$ to VSD state $j$, $z$ is the equivalent gating charge, $F$ is the Faraday constant (96485 C mol$^{-1}$), $V$ is voltage (in mV), $R$ is the gas constant (8314 J kmol$^{-1}$ K$^{-1}$), and $T$ is the absolute temperature (293 K). The intrinsic VSD-pore transitions $k_{CO}$ and $k_{OC}$ are assumed to be voltage-independent (i.e., constant), and the $\theta$ terms explicitly represent the net effect of all VSD-pore interactions within each channel state. **b** Model simulations of KCNQ1 inactivation currents stimulated with the activation protocol (upper, voltage stepped from −80 mV holding potential to −100 to +40 mV with 10 mV increment, and then stepped to −40 mV) and a triple pulse protocol (lower, same as in Fig. 1a). The insets are the enlarged currents. The values of the parameters are as follows: $k_{RI} = 0.012$ ms$^{-1}$, $k_{IR} = 0.00063$ ms$^{-1}$, $k_{IA} = 0.0019$ ms$^{-1}$, $k_{AI} = 0.0013$ ms$^{-1}$, $z_{RI} = 0.62$ e, $z_{IR} = -0.55$ e, $z_{IA} = 0.66$ e, $z_{AI} = -0.34$ e, $k_{CO} = 0.014$ ms$^{-1}$, $k_{OC} = 11.18$ ms$^{-1}$, $\theta_{RC} = 15.52$, $\theta_{IC} = 0.17$, $\theta_{AC} = 3.03$, $\theta_{RO} = 0.031$, $\theta_{IO} = 1.36$, $\theta_{AO} = 2.11$

decaying current following the short hyperpolarization (Fig. 7b). In this model, the "$K$" parameters represent transitions intrinsic in the VSD for activation and in the pore for opening, while the "$\theta$" parameters represent the interactions between the VSD and the pore when the two domains are in different states. These interactions between different structural domains define the distinct VSD-pore coupling in the IO and AO states (described by $\theta_{IO}/\theta_{IC}$ and $\theta_{AO}/\theta_{AC}$, respectively). The parameters obtained by fitting to the experimental data show that the AO/AC equilibrium is less favored than the IO/IC equilibrium, giving rise to the inactivation phenotype when the VSD activates from the intermediate state to the activated state. Since each KCNQ1 channel contains four VSDs, this model greatly simplifies the gating mechanism. A full gating model containing 30 states was established in our previous study[22]. It is remarkable that this simple kinetic model simulates the results well, supporting the concept of distinctive mechanisms for the VSD-pore coupling in the IO and AO states and that IO–AO transitions account for the inactivation in KCNQ1 channels. However, this simple conceptual model may not account for all the details of channel gating or mutational effects on gating mechanisms.

Kinetic models containing two distinct open states and one non-conducting inactivated state were previously proposed to describe KCNQ1 inactivation[33, 36, 37]. The transitions between the second open state and the inactivated state were modeled as being voltage independent, which allowed partial inactivation. Using different [Na$^+$]$_i$ sensitivities, Pusch et al.[55] demonstrated the presence of two open states of the KCNQ1 channel. Our recent studies find direct evidence for the two open states in the KCNQ1 channel, IO and AO, which are associated with the intermediate and activated state of the VSD, respectively[22, 23]. Our current study shows that the inactivation phenotype is produced directly by the distinct VSD-pore coupling in the IO and AO states, without a non-conducting inactivated state.

Some mutations of KCNQ1 such as the LQT-associated mutations S338W and L273F, result in "enhanced inactivation" with pronounced current decline during depolarization[56, 57] (Supplementary Fig. 2a). We hypothesize that the S338W and L273F mutations induce an additional inactivation in the channel that has a different mechanism than the native inactivation in our study. To test this hypothesis, we recorded the fast decaying and hook currents of the L273F mutant channel with the same voltage protocol as shown in Fig. 1a, b. Fast decaying current and hook current were observed similarly as in the WT KCNQ1 channel (Supplementary Fig. 2b), indicating that the additional inactivation introduced by the mutation does not affect the kinetics of the

native inactivation. In addition, the recovery from inactivation of L273F channels shows unique two phases, a fast phase that is similar to the WT KCNQ1 recovery from the native inactivation, and an additional slow phase that may indicate the recovery from the additional inactivation due to the very slow time course (Supplementary Fig. 2c). We also observed a steady state inactivation curve for L273F (Supplementary Fig. 2d), which shows different voltage dependence from that of the native inactivation or the activation to the AO state (Figs. 1–3). All these results indicate that the L273F mutation induces an additional inactivation in KCNQ1. Careful readers may find that the S338F channel shows some current decline at high voltages as well (Fig. 4c), which suggests that besides suppressing AO, the S338F mutation may also induce an additional inactivation in the channel.

Although step-wise activation of the VSD has been previously shown in *Shaker* K$^+$ channels[58–60], the pore opens only in the activated state of the VSD. The KCNQ1 pore opens in both the intermediate and activated states of the VSD to generate the IO and AO states, which is a unique voltage-dependent gating mechanism. The IO and AO states exhibit differences not only in voltage dependence and time course of opening but also in ion permeation and pharmacological properties[22, 23]. These differences are the basis of drastic changes in channel properties when the auxiliary KCNE1 subunits that are physiologically important are associated[22, 23]. This study shows that the elimination of inactivation by KCNE1 association is also based on the two open states, i.e., KCNE1 association suppresses the IO state to eliminate inactivation (Figs. 1 and 3). We determined, within the experimental limitations of our single-channel measurements, that the IO and AO states had similar single-channel conductances of ~0.18 pS (Fig. 6a, c). The single-channel conductance of KCNQ1 channels is consistent with previous estimates by noise analysis[49, 61, 62]. Also consistent with the results of noise analyses, we found that the association of the auxiliary subunit, KCNE1, increased the single-channel conductance of the channel (Fig. 6d). Our study shows that mutations at different sites of the KCNQ1 channel specifically suppress the IO and AO states, respectively (Figs. 3 and 4) and decrease the VSD-pore coupling in the AO state (Fig. 5). This work also shows that the open probability of the channel becomes smaller due to the switch of VSD-pore coupling from IO to AO (Fig. 6). These results are consistent with a mechanism in which the VSD-pore coupling leading to the AO state differs from that leading to the IO state. Since the IO state is specifically suppressed by a mutation that is located in the interface between the S4–S5 loop and the cytosolic side of S6[15]

(Fig. 3), the conserved interactions between the S4–S5 loop and the cytosolic side of S6 in Kv channels may couple VSD activation to pore opening in the IO state. On the other hand, a non-canonical mechanism is likely to couple VSD activation to pore opening in the AO state.

## Methods

**Constructs and mutagenesis**. Point mutations of the KCNQ1 channel were engineered using overlap extension and high-fidelity PCR. All primer sequences used in this study can be found in Supplementary Table 1. Each mutation was verified by DNA sequencing. The cRNA of mutants was synthesized using the mMessage T7 polymerase kit (Applied Biosystems-Thermo Fisher Scientific).

**Oocyte expression**. Stage V or VI oocytes were obtained from *Xenopus laevis* by laparotomy. All procedures were performed in accordance with the protocol approved by the Washington University Animal Studies Committee (protocol #20130060). Oocytes were digested by collagenase (0.5 mg/ml, Sigma-Aldrich, St Louis, MO) and injected with channel cRNAs (Drummond Nanoject, Broomall). Each oocyte was injected with cRNAs (9.2 ng) of KCNQ1 or mutations with or without KCNE1 cRNA (2.3 ng). Injected cells were incubated in ND96 solution (in mM): 96 NaCl, 2 KCl, 1.8 CaCl₂, 1 MgCl₂, 5 HEPES, 2.5 CH₃COCO₂Na, 1:100 Pen-Strep, pH 7.6) at 18 °C for at least 2 days before recording.

**Two-electrode voltage clamp**. Microelectrodes (Sutter Instrument, Item #: B150–117–10) were made with a puller (Sutter Instrument, P-97), and the resistances were 0.5–3 MΩ when filled with 3 M KCl solution. Ionic currents were recorded by Two-electrode voltage clamp (TEVC) in ND96 or 100K⁺ (NaCl was replaced with 98 mM KCl) bath solutions. Whole-oocyte currents were recorded using a CA-1B amplifier (Dagan, Minneapolis, MN) or a GeneClamp 500B amplifier (Axon Instruments, Foster City, CA) with Patchmaster (HEKA) software. The currents were sampled at 1 kHz and low-pass-filtered at 1 kHz. All recordings were carried out at room temperature (21–23 °C).

**Voltage-clamp fluorometry**. All cRNA amounts were doubled for VCF experiments in order to get higher expression level. Oocytes were incubated for 30 min on ice in 10 μM Alexa 488 C5-maleimide (Molecular Probes, Eugene, OR) in high K⁺ solution in mM (98 KCl, 1.8 CaCl₂, 5 HEPES, pH 7.6) for labeling. Cells were washed three times with ND96 solution to remove the labeling solution, and recordings were performed in ND96 solution on the CA-1B amplifier setup. Excitation and emission lights were filtered by a FITC filter cube (Leica, Germany, for Alexa 488) and the fluorescence signals were collected by a Pin20A photodiode (OSI Optoelectronics). The signals were then amplified by an EPC10 (HEKA, analog filtered at 200 Hz, sampled at 1 kHz) patch clamp amplifier and controlled by the CA-1B amplifier to make sure fluorescence signals were recorded simultaneously with currents. All other chemicals were from Sigma-Aldrich.

**Single-channel recordings of KCNQ1 and mutant channels**. Channels were expressed in mouse fibroblast ltk- cells, methods as we have previously described[63]. Cells were cultured on 25 mm glass coverslips and pieces of the coverslips with adherent cells were placed in a custom chamber containing 200 μL bath solution at room temperature (20–22 °C). Cell-attached patch single-channel currents were recorded using an Axopatch 200B patch clamp amplifier and pClamp 10 software (Molecular Devices Inc.). Borosilicate patch electrodes (Sutter Instruments, Novato, CA) had resistances of 20–60 MΩ after fire-polishing[52]. The pipette filling solution was (in mM) 135 NaCl, 5 KCl, 10 Hepes, 1 MgCl₂, 1 CaCl₂ and was adjusted to pH 7.4 with NaOH. The bath solution contained (in mM) 135 KCl, 1 MgCl₂, 1 CaCl₂, 10 HEPES and was adjusted to pH 7.4 with KOH. Single-channel currents were low-pass filtered at 2 kHz (−3 dB, four-pole Bessel filter) and sampled at 10 kHz. Further, KCNQ1, but not the mutants, was C-terminally tagged with GFP to increase the probability of recording from channel-expressing cells.

The E1R/R2E mutant appeared to be toxic to the cells, so several recordings from this mutant were obtained by adding 100 nM I(Ks) blocker HMR1556 (Tocris Bioscience, Minneapolis, MN) to the culture media of 5 h transfected cells for an overnight period. Pieces of cover glass containing cells were then transferred to fresh media without HMR1556 30 min prior to recording. We observed complete washout of the HMR1556 blocker within 10 min of whole cell recording.

**Data analysis**. Data were analyzed with IGOR (Wavemetrics, Lake Oswego, OR), Clampfit (Axon Instruments, Inc., Sunnyvale, CA), and Sigmaplot (SPSS, Inc., San Jose, CA) software. The instantaneous tail currents following test pulses were normalized to the maximal current for obtaining the conductance–voltage (G–V) relationship. Because of photo-bleaching, fluorescence signals were subtracted with a baseline that was extrapolated by fitting the first 2 s signals at the holding potential. $\Delta F/F$ was calculated after baseline subtraction. Fluorescence-voltage relationships (F–V) were derived by normalizing the $\Delta F/F$ value at the end of each four seconds test pulse to the maximal value. F–V and G–V curves were fitted with either one or the sum of two Boltzmann equations in the form $1/(1 + \exp(-z \times F \times$

$(V – V_{1/2})/RT))$ where $z$ is the equivalent valence of the transition, $V_{1/2}$ is the voltage at which the transition is half maximal, $R$ is the gas constant, $T$ is absolute temperature, $F$ is the Faraday constant, and $V$ is the voltage. For analyzing the hook currents (Fig. 1) and the AO state currents (Fig. 2), since both the activation and tail currents were generated with transitions among multiple states (Fig. 2b), the simplified description of the two open states or inactivation using double exponential fittings might introduce errors that create the small difference in the two curves.

**Data availability**. The data that support the findings of this study are available from the corresponding author upon reasonable request.

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

## Acknowledgements

We thank Dr S. Goldstein (Brandeis University) for the KCNQ1 clone, and Dr N. Schmitt for the GFP-tagged KCNQ1. We thank Dr M. Zaydman for initiating the study and helpful discussions during manuscript writing, and Professor James Ballard for proofreading of the manuscript. We thank Dr F. Xiao for the valuable suggestions on channel modeling. This work was supported by National Institutes of Health Grants R01 NS092570 and R01HL126774 to J.C. and grant number G-14-0006091 from the Heart and Stroke Foundation of Canada to D.F.

## Author contributions

P.H., conception and design of experiments and kinetic models; acquisition, analysis and interpretation of data; drafting and revising the manuscript. J.E. and D.F., conception and design of single-channel experiments; acquisition, analysis and interpretation of data; drafting and revising the manuscript. J.S. L.Z., and K.M., acquisition of molecular biology data; Y.G., acquisition of model fits; J.C., conception and design of the study; analysis and interpretation of data; drafting and revising the manuscript.

## Additional information

**Competing interests:** The authors declare no competing financial interests.

