## [Peer Review File · Nature Communications]

Reviewers' comments:

Reviewer #1 (Remarks to the Author):

The authors here address the question mechanism underlying the inactivation of KCNQ1 channels. The inactivation of KCNQ1 channels seems different from other types of inactivation, such as N- or C-type inactivation in other K channels. They here propose a novel mechanism in that the KCNQ1 inactivation is due to different types of voltage sensing domain (VSD)-pore coupling in different activated states of the VSD. First they show using VCF that the voltage dependence and time course of the inactivation and a second S4 movement, which previously was suggested to track transitions between intermediate and activated conformations of S4, are correlated, suggesting that inactivation is linked to the intermediate open (IO) to activated open(AO) transition. They further show that mutations that affect the IO to AO transition also modify inactivation. They further test if inactivation is due to changes in single channel conductance or open probability. The idea that inactivation in KCNQ1 is due to changes in coupling is novel and would be of interest for a wide range of readers. The data is of high quality and the conclusions are appropriate. However, I would like some more nuanced discussion with more alternative explanations provided (see comments below).

Major comments.

1. Page 10. S338F inactivates at higher voltages. What is this? Other studies has suggested that highly homologous S338W mutation increases inactivation (Panaghie et al 2006). Maybe some comments on this? Also, there seems to be a slow second component in the S338F currents too. What is this?
2. Pg. 11. Line 232. "reveal that the VSD-pore coupling in the AO state is distinct from that in the IO state". This seems too strong of a conclusion for the data presented. What are alternative explanations?
3. The single channel experiments are great, but can the authors really distinguish such small differences in single-channel conductance? The inactivation of the macroscopic currents are really minor (10-20% in Fig 1a). Can you detect a 10-20% change in single-channel conductance for a 0.18-pS channel? If not, then maybe there is also a change in single-channel conductance.
4. Also, what about the relatively large noise in these recordings? Isn't the single channel conductance reduced by the flickery appearance of the currents? One should really try and filter more (post recording) and see what happens if you filter at lower frequencies, to make sure that your filtering is not distorting the amplitude estimates. Not sure if this is a major problem or not.
5. Pg. 13. The change in P_o (50%) seems more than what the inactivation looks like in macroscopic recordings (10-20%). Why is that? Also figure 6f and 6g don't look too similar to me. Maybe this is just a display problem? Please indicate zero current in each panel.
6. Pg. 13. Last paragraph of Results. I don't understand this section. KCNQ1/KCNE1, which supposedly open into the AO state, has a 10 bigger single channel conductance, so what makes the currents of the AO state smaller than IO then in KCNQ1? Or is this an additional effect of KCNE1 on the currents?
7. Discussion. The conclusion that the inactivation is due to a switch in coupling between VSD and the pore seems too strong. Alternative explanations should also be given, such as, e.g., a flicker closed state being more prominent in AO than IO.

Minor Comments.

1. Pg. 3. Some references seems missing. E.g. statements on Lines 44-48 need some references.
2. Pg. 3. "decaying current following a brief hyperpolarization during a depolarizing voltage pulse" is probably better as "decaying current during a depolarizing voltage pulse following a brief hyperpolarization".
3. Pg. 9. I don't understand "These results support the idea that the inactivation in KCNQ1 derives from both IC/IO and AC/AO transitions,". Isn't the idea that inactivation occurs because of the I to A transition?
4. Pg. 11. "Voltage dependence of a state" is not a correct expression. The occupancy of a state

can be voltage dependent or the transition into/out of a state can be voltage dependent.

5. Supplementary Fig 2. This figure is hard to understand and maybe not very informative. Also the model that it is based on is not very well developed here. For example, "These results suggest that during hyperpolarization, the VSD of a fraction of the channel population moved from the activated state to the intermediate and the resting state, while some of the channels moved from the AO state to the IO and closed states (Fig. 7a)". Do you really see a closing of the channels during this protocol? It seems like the current never goes below the initial level, so this seems like a speculation. Also, "the VSD moved back to the intermediate and activated states predominately at a relatively slow rate, while the channels moved back from the IO to the AO state at a relatively fast rate." Isn't this contradicting? The transition from IO to AO is the same transition as VSD intermediate to activated, so how can it be fast in one case and slow in the other? And the transition from resting to intermediate VSD is supposedly fast, so it cannot be what cause the slow fluorescence. And not much (if any) current is decreased, so it cannot easily be transitions between closed states that generate the slow fluorescence? Maybe leave this figure out?

6. Pg. 30. Line 546. "relative fluorescent intensity of resting states" should be "activated states".

Reviewer #2 (Remarks to the Author):

The authors aimed at elucidation of the biophysical mechanisms of the inactivation of KCNQ channel, and explained it in relation to the two open states known for KCNQ channels, IO (intermediate) and AO (full). They observed that the inactivating component seen as a hook in the tail current (A2) correlates well with the slowly activating current component upon depolarization (Islow). Next, they analyzed the properties of F351A mutant in which IO is eliminated, S338F mutant whose AO is lost, and D242N mutant in which the transition to AO is decreased. Based on the results, they concluded that the inactivation is due to the transition, from IO with higher VSD-pore coupling, to AO with lower coupling. They also analyzed single channel properties and showed that the single channel conductance is not significantly different between IO and AO, but that the activity of IO is higher than that of AO. Finally they performed simulation analyses using kinetic model based on their experimental results and reproduced inactivation successfully.

The novel finding that the inactivation of KCNQ channel is due to the difference of the activities between the two open states has a high impact to broad ranged readers. I believe the scientific merit is worthy for publication in a prestigious journal such as Nature Communications. The experiments are carefully designed and well performed. The data quality is generally high and the manuscript is written mostly well.

I have some specific comments which require attention. I would like to ask the authors for clarifications as well as revisions.

1. Fig. 1d vs Fig. 1e

Time course of A2 (also A1) looks very different in these two plots, almost ten times. Which is correct?

2. Fig. 2d

I understand that +40 mV and -20 mV are membrane potentials where Islow was recorded, whereas they are prepulse potentials for A2, and the tail (hook) current was recorded at -120 mV. I cannot fully understand why the time courses of A2 and Islow recorded at different potentials are similar. (Am I misunderstanding something?)

3. Fig. 4b and Fig. 4e

(1) In Fig. 4b recorded up to +60 mV, there is a slowly declining component, although a hook is not observed in Fig. 4a. What does this declining component mean?

(2) Also, in Fig 4e (left) recorded up to +100 mV, why wasn't the declining component seen in Fig.

4b reproduced?

4. Fig. 4e (right)

(1) Why there is no F2 here? If AC is intact, I think F2 could be observed. Is it possible not only AO but also AC is lost in S338F mutant?

(2) Voltage label of X-axis is missing.

5. Fig. 5c,d

This results of D242N mutant here are quite puzzling. I understand F2 is due to the transition from I state to A state. I cannot understand well why F2 is left shifted, when A2 and IAO are right shifted. I think the explanation in the text is not sufficient.

6. Fig. 6a,c

These results clearly demonstrate that single channel conductance of E1R/R2E (Fig. 6a) and E1R/R4E (Fig. 6C) are similar. However, isn't there a possibility that the single channel conductance of E1R/R2E (E1R/R4E) does not necessarily reflect that in IO of wt (or AO of wt)?

7. Fig 6f, g

(1) The authors described that the ensemble average of single channel (Fig. 6f) and macropatch (Fig. 6g) current recordings showed decline due to the decrease in Popen during the sustained depolarizing potential caused by the transition from IO to AO. The authors wrote the decline was observed also in the whole cell patch current recording by repeated pulses (Fig. 6g). However, it is not observed in the macroscopic current from oocytes (e.g. Fig. 1b, c, Fig. 2c) during the sustained depolarization pulse. I would like to ask for clarification.

(2) If IIO is activated rapidly to a peak level, and if the same channel moves from the IIO to the IAO state with less Popen as explained in the single channel recording part, why macroscopic IIO and IAO look as shown Fig. 2C?

Why there is no decline of IIO (and of whole macro current) due to the decrease of the fraction?

(3) In sum, I cannot fully understand the relationship of Fig. 6e,f,g and the rest of the paper.

8. Fig. 7

(1) Why there is no current decline observed in ensemble average of single channel recording?

(2) Do the parameters fit well with the very high Popen in the single channel recording?

(3) Can FV (fraction of I and A) predicted from this simulation? Isn't it worthy for presenting?

(4) What will happen, if IO state or AO state is eliminated in the simulation? Does the simulation reproduce the results of the mutants? Isn't it worthy for presenting, even if it is not very qualitative?

Reviewer #3 (Remarks to the Author):

Thank you for the opportunity to review this interesting paper from Hou et al., entitled 'Inactivation of KCNQ1 potassium channels reveals dynamic couplings between voltage sensing and pore opening'. The paper rationalizes gating properties of KCNQ1 referred to as inactivation (and shown to be prevented by association with the auxiliary subunit KCNE1). The authors provide compelling evidence that the apparent inactivation of KCNQ1 (revealed by brief hyperpolarizations that lead to enhanced currents in a subsequent depolarization) arises from changes in distribution between two distinct open states: 'Activated' (weaker VSD-pore coupling) vs. 'Intermediate' (stronger coupling). In addition to the effects of KCNE1, the authors present several mutations that weaken the appearance of inactivation by either preventing the appearance of the Intermediate open state (F351A), or the Activated open state (S338F), along with combinations of

these mutations with VSD mutations that trap the voltage sensor in either the intermediate or activated state. The approach of the combination of mutations that suppress or enhance intermediate or activated sense is complex but internally consistent and lucid. The authors also present very challenging single channel recordings from these channels with extremely low conductance. A shortcoming of the paper is that I had difficulty reconciling the outcome of the single channel and macroscopic data presented.

Overall, the topic is interesting, and there are several very interesting findings described here. However, there were some inconsistencies that I would like the authors to address (I am unsure whether these reflect my lack of understanding of their model, but it is unlikely I would not be the only reader to struggle with these issues).

Major comments:

1. It was not clear why the outcome of the KCNQ1 ensemble average (or multi-channel patch) was different from the macroscopic currents shown earlier. The authors point out that the ensemble average shows a progressive reduction of open probability (attributed to a switch from IO to AO states). However this is distinct from the currents shown for KCNQ1 in Figure 2 (which had a slow activating phase attributed to recruitment of channels to AO).

2. More generally, I struggled to understand how redistribution between the IO/IC and AO/AC gating equilibria could lead to an increase in current as shown in Figure 1,2. Presumably, a shift of channels from IO/IC (rapidly activating, higher P_o) into AO/AC (slow onset, lower P_o) should cause a reduction of current. This should be true especially at the positive voltages used, where the F1-V relationship is saturated (so a channel gained in AO/AC means a channel is lost from IO/IC). I would be grateful if the authors could explain this in more detail as this does not seem consistent.

Minor comments:

Figure 1e and other similar plots. The X-axis meaning could be clarified by labeling as 'prepulse duration'?

It is not clear how the protocols differ between Figure 2a and c (tail currents are different, 2c is lacking a time scale)

Figure 4 was difficult to follow because the panels were scattered without labels (this could be organized better for clarity).

There seems to be some variation in voltage-dependence and steepness of S338F between Fig. 4b and e. Is there a lot of variation in these parameters in different preparations or oocytes? The differences seem quite obvious in the tail currents.

Voltage axis is not labeled in Figure 4E

Legend Figure 7 has relative fluorescence intensity for different model states, but the model simulations do not show predicted fluorescence.

Reviewers' comments:

Reviewer #1 (Remarks to the Author):

The authors here address the question mechanism underlying the inactivation of KCNQ1 channels. The inactivation of KCNQ1 channels seems different from other types of inactivation, such as N- or C-type inactivation in other K channels. They here propose a novel mechanism in that the KCNQ1 inactivation is due to different types of voltage sensing domain (VSD)-pore coupling in different activated states of the VSD. First they show using VCF that the voltage dependence and time course of the inactivation and a second S4 movement, which previously was suggested to track transitions between intermediate and activated conformations of S4, are correlated, suggesting that inactivation is linked to the intermediate open (IO) to activated open(AO) transition. They further show that mutations that affect the IO to AO transition also modify inactivation. They further test if inactivation is due to changes in single channel conductance or open probability. The idea that inactivation in KCNQ1 is due to changes in coupling is novel and would be of interest for a wide range of readers. The data is of high quality and the conclusions are appropriate. However, I would like some more nuanced discussion with more alternative explanations provided (see comments below).

We very much appreciate the time and effort the reviewer has put into the insightful comments.

Major comments.

1. Page 10. S338F inactivates at higher voltages. What is this? Other studies has suggested that highly homologous S338W mutation increases inactivation (Panaghie et al 2006). Maybe some comments on this?

The S338F channel shows current decline at high voltages (currents without KCNE1 in Fig 4c), which is consistent with previous studies [Michael Hoosien et al. Heart Rhythm. 2013]. One possibility is that besides suppressing AO, the S338F mutation induces an additional inactivation in the channel. To demonstrate this, we tested a LQT mutation, L273F, that shows “enhanced inactivation”, just as the S338W and F339W channels show in Panaghie et al 2006. Although the L273F channel shows “enhanced inactivation”, a fast decaying current and hook current are observed just as in WT KCNQ1 channels. Our experimental data support that L273F induces an additional inactivation in the channel, which are shown in a new supplementary figure (Supplementary Fig 2). We describe the L273F results in the Discussion. We also discuss the S338F results in the same paragraph (please see pg. 15-16).

Also, there seems to be a slow second component in the S338F currents too. What is this?

The slow second component in the S338F currents (currents in Fig. 4f) only shows up when S338F is in the VCF background mutations of KCNQ1 (C214A/G219C/C331A). These VCF mutations make the S338F currents smaller such that the currents are easier to be contaminated by endogenous slow currents. In addition, these VCF mutations may alter channel function by bringing in additional kinetic components. Actually, in WT KCNQ1 these VCF mutations change the GV relationship and gating kinetics (Osteen et al.

PNAS. 2010). Because of these unexpected changes, we always use currents of the WT or mutations on the WT background for analyzing the inactivation properties.

2. Pg. 11. Line 232. “reveal that the VSD-pore coupling in the AO state is distinct from that in the IO state”. This seems too strong of a conclusion for the data presented. What are alternative explanations?

We agree with the reviewer’s comment. One alternative we can think of could be that the coupling mechanisms for the IO and AO states may differ but share some common structures or interactions. We change “reveal that the VSD-pore coupling in the AO state is distinct from that in the IO state.” into “reveal that VSD-pore coupling in the IO and AO states is different and can be altered separately by distinctive mutations” in the revised manuscript (please see pg. 11).

3. The single channel experiments are great, but can the authors really distinguish such small differences in single-channel conductance? The inactivation of the macroscopic currents are really minor (10-20% in Fig 1a). Can you detect a 10-20% change in single-channel conductance for a 0.18-pS channel? If not, then maybe there is also a change in single-channel conductance.

We agree with the reviewer that recording and analyzing such small single channel currents are challenging. So, although average data did not reveal a difference in single channel conductance between E1R/R2E and E1R/R4E, we have now modified our statement in the Discussion to say that (pg. 16):

We determined, within the experimental limitations of our single channel measurements, that the IO and AO states had similar single channel conductances of ~0.18 pS (Fig. 6a,c).

4. Also, what about the relatively large noise in these recordings? Isn't the single channel conductance reduced by the flickery appearance of the currents? One should really try and filter more (post recording) and see what happens if you filter at lower frequencies, to make sure that your filtering is not distorting the amplitude estimates. Not sure if this is a major problem or not.

We know that the conductance of KCNQ1 is dependent on the filter frequency (Yang and Sigworth, 1998), and so the aim in our experiments was to ensure that all data were filtered at 200 Hz in a similar manner post-collection. When we filtered data at 200 and 500 Hz in a recent study (Thompson et al, 2017 JGP Aug 7;149(8):781-798.), conductance was increased by ~10%, but remained very similar between constructs.

5. Pg. 13. The change in P_o (50%) seems more than what the inactivation looks like in macroscopic recordings (10-20%). Why is that?

In the single channel data the ensemble average was only obtained from active sweeps (as now stated in the legend, pg. 30), and this, of course will greatly increase the apparent inactivation as active channels transition from the IO to AO states. If all the blank sweeps were included (for Q1, ~90% of sweeps are blank) the effect of the active sweeps would be difficult to discern and the degree of inactivation would appear to be much less (see figure below). We now emphasize the condition of this analysis in the text.

Ensemble Average of all useable sweeps (n=75)

Also figure 6f and 6g don't look too similar to me. Maybe this is just a display problem? Please indicate zero current in each panel.

The reviewer is correct and thanks for catching the differences. Macropatch currents varied between cells and the change in current amplitude is not as prominent as in the ensemble average of the active traces of single channel records. We have now omitted these data to allow us to expand the ensemble average. Zero currents are now shown in all panels of Fig. 6.

6. Pg. 13. Last paragraph of Results. I don't understand this section. KCNQ1/KCNE1, which supposedly open into the AO state, has a 10 bigger single channel conductance, so what makes the currents of the AO state smaller than IO then in KCNQ1? Or is this an additional effect of KCNE1 on the currents?

In KCNQ1 alone, the IO and AO states have the same single channel conductance (Fig 6a, c, e). The reviewer is right that KCNE1 has an additional effect on the single channel conductance, which is to increase single channel conductance (Yang Y, Sigworth FJ. Single-Channel Properties of I_{Ks} Potassium Channels. *The Journal of General Physiology*. 1998;112(6):665-678.; Murray et al. Unnatural amino acid photo-crosslinking of the I_{Ks} channel complex demonstrates a KCNE1:KCNQ1 stoichiometry of up to 4:4. *eLife*. 2016) (Fig 6d). Since the channel associated with KCNE1 only opens to the AO state, KCNE1 increases the single channel conductance of the AO state. However, in the absence of KCNE1, the AO state single channel conductance is the same as in the IO state. We now point out this effect of KCNE1 to clarify these issues.

7. Discussion. The conclusion that the inactivation is due to a switch in coupling between VSD and the pore seems too strong. Alternative explanations should also be given, such as, e.g., a flicker closed state being more prominent in AO than IO.

We appreciate the argument of the reviewer on how AO differs from IO, whether the difference is due to a more prominent flicker closed state or a reduced open probability by another mechanism. However, regardless how AO differs from IO, we observe that inactivation happens only when the channel enters the AO state and makes transitions between the IO and AO states. Our results also show that the AO state is coupled to the activated state of the VSD, while the IO state is coupled to the intermediate state

of the VSD, and the two coupling mechanisms are different. These results lead to the conclusion that the inactivation in KCNQ1 is the manifestation of the switch in coupling between VSD and the pore. We revised the first paragraph of Discussion to make the derivation of our conclusion clearer (pg. 14).

Minor Comments.

1. Pg. 3. Some references seem missing. E.g. statements on Lines 44-48 need some references.

We thank the reviewer for the careful review. We added references as suggested (pg. 3).

2. Pg. 3. “decaying current following a brief hyperpolarization during a depolarizing voltage pulse” is probably better as “decaying current during a depolarizing voltage pulse following a brief hyperpolarization”.

We are trying to emphasize that the brief hyperpolarization is in the middle of a depolarizing pulse, and we feel that this emphasis is lost in the reviewer’s suggestion. We therefore respectfully prefer not to make the change.

3. Pg. 9. I don’t understand “These results support the idea that the inactivation in KCNQ1 derives from both IC/IO and AC/AO transitions,”. Isn’t the idea that inactivation occurs because of the I to A transition?

We realize that the expression “IC/IO and AC/AO transitions” is perhaps poorly worded. We have now changed this to “These results support the idea that the inactivation in KCNQ1 derives from the transitions between IO and AO states” (pg. 7,8,9,13).

4. Pg. 11. “Voltage dependence of a state” is not a correct expression. The occupancy of a state can be voltage dependent or the transition into/out of a state can be voltage dependent.

We are now using “voltage dependent activation to the AO state” (pg. 11).

5. Supplementary Fig 2. This figure is hard to understand and maybe not very informative. Also the model that it is based on is not very well developed here. For example, “These results suggest that during hyperpolarization, the VSD of a fraction of the channel population moved from the activated state to the intermediate and the resting state, while some of the channels moved from the AO state to the IO and closed states (Fig. 7a)”. Do you really see a closing of the channels during this protocol? It seems like the current never goes below the initial level, so this seems like a speculation. Also, “the VSD moved back to the intermediate and activated states predominately at a relatively slow rate, while the channels moved back from the IO to the AO state at a relatively fast rate.” Isn’t this contradicting? The transition from IO to AO is the same transition as VSD intermediate to activated, so how can it be fast in one case and slow in the other? And the transition from resting to intermediate VSD is supposedly fast, so it cannot be what cause the slow fluorescence. And not much (if any) current is decreased, so it

cannot easily be transitions between closed states that generate the slow fluorescence? Maybe leave this figure out?

We think that the reviewer's suggestion to leave this figure out is good. The figure is not necessary to the main conclusion and leaving it out does not affect the manuscript as whole.

6. Pg. 30. Line 546. "relative fluorescent intensity of resting states" should be "activated states".

Thanks for catching this mistake. Corrected (pg. 31).

Reviewer #2 (Remarks to the Author):

The authors aimed at elucidation of the biophysical mechanisms of the inactivation of KCNQ channel, and explained it in relation to the two open states known for KCNQ channels, IO (intermediate) and AO (full). They observed that the inactivating component seen as a hook in the tail current (A2) correlates well with the slowly activating current component upon depolarization (I_{slow}). Next, they analyzed the properties of F351A mutant in which IO is eliminated, S338F mutant whose AO is lost, and D242N mutant in which the transition to AO is decreased. Based on the results, they concluded that the inactivation is due to the transition, from IO with higher VSD-pore coupling, to AO with lower coupling. They also analyzed single channel properties and showed that the single channel conductance is not significantly different between IO and AO, but that the activity of IO is higher than that of AO. Finally they performed simulation analyses using kinetic model based on their experimental results and reproduced inactivation successfully.

The novel finding that the inactivation of KCNQ channel is due to the difference of the activities between the two open states has a high impact to broad ranged readers. I believe the scientific merit is worthy for publication in a prestigious journal such as Nature Communications. The experiments are carefully designed and well performed. The data quality is generally high and the manuscript is written mostly well.

I have some specific comments which require attention. I would like to ask the authors for clarifications as well as revisions.

We appreciate the positive comments, thanks.

1. Fig. 1d vs Fig. 1e

Time course of A2 (also A1) looks very different in these two plots, almost ten times. Which is correct?

Both are correct. The time scale in Fig. 1d indicates the time dependence of the hook current, but the time scale in Fig. 1e is the time duration of the pre-pulse. We have changed the axis label in Fig 1e for clarification.

2. Fig. 2d

I understand that +40 mV and -20 mV are membrane potentials where I_{slow} was recorded, whereas they are prepulse potentials for A2, and the tail (hook) current was recorded at -120 mV. I cannot fully understand why the time courses of A2 and I_{slow} recorded at different potentials are similar. (Am I misunderstanding something?)

Both A_2 and I_{slow} reflect the AO state, and that's why they have similar time dependence. +40 mV and -20 mV in Fig. 2c,d are voltages where I_{slow} was recorded, and this is a direct measurement of the AO state. +40 mV and -20 mV in Fig.1 b,e are prepulse voltages that develop AO to different time durations, then -120 mV is used to record the tail (hook) currents to measure the inactivation. So the inactivation is an indirect measurement of the AO state during the prepulses at +40 and -20 mV. Now with the change of the axis label in Fig 1e we hope these results are less confusing.

3. Fig. 4b and Fig. 4e.

(1) In Fig. 4b recorded up to +60 mV, there is a slowly declining component, although a hook is not observed in Fig. 4a. What does this declining component mean?

Please see the response to Reviewer 1's Major Comment 1.

(2) Also, in Fig 4e (left) recorded up to +100 mV, why wasn't the declining component seen in Fig. 4b reproduced?

We think that this is because the VCF background mutations (C214A/G219C/C331A) may alter channel function to make the behavior different from the channel without these mutations (Osteen et al. PNAS. 2010). We now indicate the background constructs for mutations in all figures.

4. Fig. 4e (right)

(1) Why there is no F2 here? If AC is intact, I think F2 could be observed. Is it possible not only AO but also AC is lost in S338F mutant?

The original figure has F2 plotted as a dashed line and labeled.

(2) Voltage label of X-axis is missing.

Thanks for pointing this out. Corrected.

5. Fig. 5c,d

This results of D242N mutant here are quite puzzling. I understand F2 is due to the transition from I state to A state. I cannot understand well why F2 is left shifted, when A2 and IAO are right shifted. I think the explanation in the text is not sufficient.

The results suggest that the mutation D242N reduces the coupling between VSD and the pore such that the pore is opened less efficiently and requires higher voltages to open, while the VSD has less load during activation and requires lower voltages to activate. This was nicely demonstrated in the paper by Chowdhury and Chanda (2012) (ref 47). We have added a similar explanation in the text (pg. 11).

6. Fig. 6a,c

These results clearly demonstrate that single channel conductance of E1R/R2E (Fig. 6a) and E1R/R4E (Fig. 6C) are similar. However, isn't there a possibility that the single channel conductance of E1R/R2E (E1R/R4E) does not necessarily reflect that in IO of wt (or AO of wt)?

We agree with the reviewer that it is possible that the single channel conductances of E1R/R2E and E1R/R4E, which are mutant channels, may not necessarily reflect that of IO or AO in WT channels. However, the similarity between the conductances of E1R/R2E and E1R/R4E and KCNQ1 suggest that these mutations, which affect VSD position, do not alter pore conductance appreciably. As well, when co-expressed with KCNE1 (Fig. 6D) the conductances of E1R/R4E, and WT KCNQ1+KCNE1 are very similar.

7. Fig 6f, g

(1) The authors described that the ensemble average of single channel (Fig. 6f) and macropatch (Fig. 6g) current recordings showed decline due to the decrease in Popen during the sustained depolarizing potential caused by the transition from IO to AO. The authors wrote the decline was observed also in the whole cell patch current recording by repeated pulses (Fig. 6g). However, it is not observed in the macroscopic current from oocytes (e.g. Fig. 1b, c, Fig. 2c) during the sustained depolarization pulse. I would like to ask for clarification.

Please see our response to the Major Comment 5 of Reviewer 1 with regarding to the whole cell patch current.

On the comparison between KCNQ1 currents in mammalian cells and in oocytes, it is known that the KCNQ1 channel current kinetics recorded in mammalian cells is different from that in oocytes. Mammalian KCNQ1 channels shows a decline in current at high voltages (Fig. 6f,g) (Chen et al. Pacing Clin Electrophysiol. 2011) while oocyte KCNQ1 an increase (Fig. 2). Currents in oocytes are generally slower to activate than in mammalian cells, and this slow activation overlaps the transition from IO to AO, so that inactivation is best seen in the hooked tail currents during recovery.

The KCNQ1 channel is activated by not only voltage but also intracellular signaling molecules such as PIP_2 and ATP. Different cell types may have different concentrations of these signaling molecules which can alter KCNQ1 kinetics. The detailed reason needs to be further studied.

(2) If IIO is activated rapidly to a peak level, and if the same channel moves from the IIO to the IAO state with less P_{open} as explained in the single channel recording part, why macroscopic IIO and IAO look as shown Fig. 2C?

Why there is no decline of IIO (and of whole macro current) due to the decrease of the fraction?

The IIO is activated rapidly to a peak level, and then AO is slowly developed during depolarization although the transitions between IO and AO are very fast. The current kinetics depends on both the rates of VSD activation and $\text{IO} \leftrightarrow \text{AO}$ transitions. When the $\text{IO} \leftrightarrow \text{AO}$ transitions are much faster than the VSD activation we expect to observe our results in Fig 2c (see Fig 7 the kinetic model and simulations).

(3) In sum, I cannot fully understand the relationship of Fig. 6e,f,g and the rest of the paper.

Fig. 6e,f (g is deleted) are KCNQ1 currents recorded in mammalian cells, which show the data supporting a transition from the IO state to the AO state with a reduced open probability. These results suggest the mechanism of why the IO-AO transition results in the inactivation-like phenotype.

8. Fig. 7

(1) Why there is no current decline observed in ensemble average of single channel recording?

The current does decline, but is difficult to see on this scale. We have enlarged and filtered the ensemble record to show the decline more clearly (Fig 6f).

(2) Do the parameters fit well with the very high P_{open} in the single channel recording?

The high P_{open} was obtained from an average of the active sweeps from a set of 75, so the overall P_{open} is much lower. We apologize for omitting this important information, and it is now included in the text to the Discussion and the legend. However, it is still unlikely that the model parameters would fit as the single channel recordings are made in mammalian cells while our model is built based on the data from recordings in oocytes. The kinetics are different in these two cell types as pointed out by the reviewer in the above comments for the reasons that may be associated with PIP_2 or ATP concentrations.

(3) Can FV (fraction of I and A) predicted from this simulation? Isn't it worthy for presenting?

It would have been nice to have a fit to both the current and VSD activation. However, we obtain the fluorescence data using a pseudo wild type channel with mutations (C214A/G219C/C331A) using VCF.

These VCF mutations alter channel properties including shifting the G-V relation and changing current kinetics (Osteen et al. PNAS. 2010). It is not clear how these mutations alter VSD activation and therefore it is hard for us to use the fluorescence data obtained in VCF to build a model of VSD activation in the WT KCNQ1.

(4) What will happen, if IO state or AO state is eliminated in the simulation? Does the simulation reproduce the results of the mutants? Isn't it worthy for presenting, even if it is not very qualitative?

The model we presented in the manuscript is a conceptual model to show that our conclusion on inactivation is thermodynamically plausible. It is a simplified model without considering many important properties of KCNQ1 channels including 1) the channel is a homotetramer of KCNQ1, while our model treats the channel as a monomer, 2) the channel activation is dependent on PIP2 and ATP, while our model does not include these mechanisms. We do not intend for this model to account for all the mechanisms of the KCNQ1 channel, and the mutational effects may cause other changes in channel activation than the inactivation phenotype, which is beyond the scope of this manuscript.

Reviewer #3 (Remarks to the Author):

Thank you for the opportunity to review this interesting paper from Hou et al., entitled 'Inactivation of KCNQ1 potassium channels reveals dynamic couplings between voltage sensing and pore opening'. The paper rationalizes gating properties of KCNQ1 referred to as inactivation (and shown to be prevented by association with the auxiliary subunit KCNE1). The authors provide compelling evidence that the apparent inactivation of KCNQ1 (revealed by brief hyperpolarizations that lead to enhanced currents in a subsequent depolarization) arises from changes in distribution between two distinct open states: 'Activated' (weaker VSD-pore coupling) vs. 'Intermediate' (stronger coupling). In addition to the effects of KCNE1, the authors present several mutations that weaken the appearance of inactivation by either preventing the appearance of the Intermediate open state (F351A), or the Activated open state (S338F), along with combinations of these mutations with VSD mutations that trap the voltage sensor in either the intermediate or activated state. The approach of the combination of mutations that suppress or enhance intermediate or activated sense is complex but internally consistent and lucid. The authors also present very challenging single channel recordings from these channels with extremely low conductance. A shortcoming of the paper is that I had difficulty reconciling the outcome of the single channel and macroscopic data presented.

Overall, the topic is interesting, and there are several very interesting findings described here. However, there were some inconsistencies that I would like the authors to address (I am unsure whether these reflect my lack of understanding of their model, but it is unlikely I would not be the only reader to struggle with these issues).

We thank the reviewer for constructive comments and suggestions.

Major comments:

1. It was not clear why the outcome of the KCNQ1 ensemble average (or multi-channel patch) was different from the macroscopic currents shown earlier. The authors point out that the ensemble average shows a progressive reduction of open probability (attributed to a switch from IO to AO states). However this is distinct from the currents shown for KCNQ1 in Figure 2 (which had a slow activating phase attributed to recruitment of channels to AO).

Please see our response to Comment 7(1) of Reviewer 2.

2. More generally, I struggled to understand how redistribution between the IO/IC and AO/AC gating equilibria could lead to an increase in current as shown in Figure 1,2. Presumably, a shift of channels from IO/IC (rapidly activating, higher P_o) into AO/AC (slow onset, lower P_o) should cause a reduction of current. This should be true especially at the positive voltages used, where the F1-V relationship is saturated (so a channel gained in AO/AC means a channel is lost from IO/IC). I would be grateful if the authors could explain this in more detail as this does not seem consistent.

Please see our response to Comment 7(2) of Reviewer 2.

Minor comments:

Figure 1e and other similar plots. The X-axis meaning could be clarified by labeling as 'prepulse duration'?

Thanks for the suggestion. Done.

It is not clear how the protocols differ between Figure 2a and c (tail currents are different, 2c is lacking a time scale)

The protocol is the same except for that the highest voltage in 2a is 100 mV but in 2c is 80 mV. We have added a description of the protocols for Fig. 2a and c in the figure legend. The a and c panels show the pseudo-WT KCNQ1 (C214A/G219C/C331A) and WT KCNQ1, respectively. The differences observed by the reviewer are due to different properties of these two channels. We now indicate the background constructs for mutations in all figures.

We have added scale bars in Fig. 2c.

Figure 4 was difficult to follow because the panels were scattered without labels (this could be organized better for clarity).

We have now labeled every panel in the figure.

There seems to be some variation in voltage-dependence and steepness of S338F between Fig. 4b and e. Is there a lot of variation in these parameters in different preparations or oocytes? The differences seem quite obvious in the tail currents.

The difference is due to different channel background. Fig. 4b is the S338F mutation in the WT KCNQ1 background, while Fig.4e is the same mutation in the pseudo-WT KCNQ1 (C214A/G219C/C331A) background. The data are very consistent within each construct.

Voltage axis is not labeled in Figure 4E

Thanks. Corrected.

Legend Figure 7 has relative fluorescence intensity for different model states, but the model simulations do not show predicted fluorescence.

Please see our response to Comment 8(3) of Reviewer 2.

REVIEWERS' COMMENTS:

Reviewer #1 (Remarks to the Author):

The authors have responded well to my comments.

Reviewer #2 (Remarks to the Author):

The authors fully answered to all my comments to the previous version, and revised the manuscript satisfactorily. I have no further comments. I evaluate this paper has good scientific merits and high general impact worthy for publication in Nature Communications in the present form.

Reviewer #3 (Remarks to the Author):

The revised article by Hou et al. entitled 'Inactivation of KCNQ1 potassium channels reveals dynamic coupling between voltage sensing and pore opening' is significantly improved in terms of clarity from the initial version and most of my comments have been addressed. These are just some small points that I believe the authors can clarify easily:

1. The legend for Figure 7 has a statement about Fluorescence intensity values of different states (last sentence of the legend). This should be removed since the figure does not involve any simulations of fluorescence intensity.
2. I think the description of the model could be significantly improved with a better explanation of the slow rising phase that is observed/simulated as increased number of channels move in the AC/AO equilibrium (there must be some reserve of channels in RC/RO that can repopulate IC/IO as these channels activate further to AC/AO). I did not find the response/explanation of this to be clear (the authors mention that the IO-AO transitions are much faster than VSD activation, but doesn't the IO-AO transition involve a voltage sensor movement?)

REVIEWERS' COMMENTS:

Reviewer #1 (Remarks to the Author):

The authors have responded well to my comments.

Thanks.

Reviewer #2 (Remarks to the Author):

The authors fully answered to all my comments to the previous version, and revised the manuscript satisfactorily. I have no further comments. I evaluate this paper has good scientific merits and high general impact worthy for publication in Nature Communications in the present form.

Thanks.

Reviewer #3 (Remarks to the Author):

The revised article by Hou et al. entitled 'Inactivation of KCNQ1 potassium channels reveals dynamic coupling between voltage sensing and pore opening' is significantly improved in terms of clarity from the initial version and most of my comments have been addressed.

These are just some small points that I believe the authors can clarify easily:

1. The legend for Figure 7 has a statement about Fluorescence intensity values of different states (last sentence of the legend). This should be removed since the figure does not involve any simulations of fluorescence intensity.

Thank you, and we have removed the statement about fluorescence intensity values of different states.

2. I think the description of the model could be significantly improved with a better explanation of the slow rising phase that is observed/simulated as increased number of channels move in the AC/AO equilibrium (there must be some reserve of channels in RC/RO that can repopulate IC/IO as these channels activate further to AC/AO). I did not find the response/explanation of this to be clear (the authors mention that the IO-AO transitions are much faster than VSD activation, but doesn't the IO-AO transition involve a voltage sensor movement?)

We appreciate the reviewer's comment on the description of the model. However, the kinetics of the model is complex, determined by different rates of many transitions. Although the time course of current increase recapitulates the

experimental data, it is difficult to simply attribute the current rising to a couple of steps without having to make many quantifications. We tried to add some interpretation with words into the manuscript, but we feel that these words are not as precise or correct as the model and parameters listed in fig. 7 can tell. We have to give up the effort and leave this part of the manuscript unchanged.